# Effect of Pre-Stretch on the Precipitation Behavior and the Mechanical Properties of 2219 Al Alloy

**DOI:** 10.3390/ma14092101

**Published:** 2021-04-21

**Authors:** Guo-Ai Li, Zheng Ma, Jian-Tang Jiang, Wen-Zhu Shao, Wei Liu, Liang Zhen

**Affiliations:** 1Beijing Institute of Aeronautical Materials, Beijing 100095, China; LLLDK3@163.com; 2Beijing Engineering Research Center of Advanced Aluminum Alloys and Application, Beijing 100095, China; 3School of Materials Science and Engineering, Harbin Institute of Technology, Harbin 150001, China; mz1081900114@163.com (Z.M.); jjtcy@hit.edu.cn (J.-T.J.); wzshao@hit.edu.cn (W.-Z.S.); liuw@hit.edu.cn (W.L.); 4National Key Laboratory of Precision Hot Processing of Metals, Harbin Institute of Technology, Harbin 150001, China

**Keywords:** Al–Cu alloy, pre-stretch, cold-work strengthening, age-hardening, tensile properties

## Abstract

The influence of pre-stretch on the mechanical properties of 2219 Al alloys sheets were systematically investigated, with the aim of examining the age-strengthening in parts draw-formed from as-quenched sheets. The precipitation was characterized based on differential scanning calorimetry (DSC) analysis and transmission electron microscope (TEM) observation of specimens of as-quenched and quenched-stretched condition to address the influence of pre-stretching. A tensile test was performed to evaluate the effect on mechanical properties. The introduction of pre-stretching endues increased yield strength (YS) and thus can be helpful to exert the potential of the alloy. Peak YS of 387.5 and 376.8 MPa are obtained when specimens pre-stretched for 10% are aged at 150 and 170 °C, respectively, much higher than that obtained in the non-stretched specimens (319.2 MPa). The precipitation of Guinier-Preston zone (G.P. zones) and the transition to *θ″* shifts to a lower temperature when pre-stretched is performed. The high density of dislocations developed during the stretching contributes to the acceleration in precipitation. Quench-stretched specimens present a much quicker age-hardening response at the beginning stage, which endue higher peaked yield strength. The yield strength, however, decrease much more quickly due to the recovery that occurs during the aging processes. The study suggested the feasibility of aging draw-formed components of 2219 Al alloy to obtain high strength.

## 1. Introduction

Aluminum alloys of Al–Cu series have been widely applied in the spacecraft industry for fabricating thin wall components including cabins, tanks, and fuselages, for their high specific stiffness, high specific strength, and the good weld ability [1,2,3,4,5]. Most thin wall components are fabricated through draw-forming in which tempered sheets are drawn against dies to form certain curves. The fabricated components then need to be put into solution-aging treatment to achieve age-strengthening. Severe distortion, however, may occur in formed components when quenched. The post forming quench-aging procedure sometimes then must be canceled to avoid the quench-induced distortion, which wastes the age-strengthening potential of Al–Cu alloys. A possible way to solve the problem is to form Al–Cu sheets of solution condition into the final shape and then age the draw-formed components to achieve high strength. The age-strengthening effect developed in the post-forming aging can be crucial for performance tailoring. For instance, the fabricating process of fuselages indicates that plastic strain up to 15%, which is much larger than that applied in the conventional prior-aging stretching, develops in the component during the draw-forming process. The large cold stretching applied during the forming, together with the subsequent aging, will then lead to increased mechanical performance in the finished fuselages. To figure out the effect of pre-stretch in a wide range is then a basic mission for designing fabricating process for thin wall components.

The precipitation sequence of Al–Cu alloys has been investigated extensively [6,7,8,9,10] and the effect of the stretch prior to aging on the precipitation has been of specific concern [11,12,13,14,15]. Small amounts (1~8%) of pre-stretch were observed to enhance the age hardening response in a few of Al–Cu–x alloys [13,14,15]. For instance, Ünlü [13] observed an enhanced age-hardening in an Al–Cu–Mg alloy both in natural and artificial aging process when 6% of pre-stretch was carried out in specimens of solution condition. Ringer [14] also observed a similar effect in pre-stretched specimens at the early stage of natural aging. It was also noted [16] that the yield strength of 2219 forgings increased by 26.2% when a 3% pre-deformation was introduced before putting into aging. The increase in strength was attributed to the combined effects of heterogeneous nucleation and growth of precipitates at dislocations due to the pre-deforming. The intensified aging-hardening is believed to root from the enhanced precipitation. It was noted that the introducing of pre-strain can stop the formation of GP zone and facilitate the precipitation of *θ’*-phases and *S’*-phase along dislocations [17]. Specifically, the interaction between gliding dislocations and Cu atoms leads to the forming of the *θ’*-phase. Liu [18] introduced deformation of 2% to Al–Cu alloy and observed an increased age-hardening response, which was associated with the accelerated heterogeneous formation of *θ’* precipitates on dislocations. The presence of high density of dislocations that developed during the deformation is believed promote the aging since they can serve as preferential matrix nucleation and growth sites for precipitates [19]. Ringer [14] observed that the application of a mechanical stretch enhanced the nucleation of *θ’* phases in Al–Cu–Mg alloy, leading to a more uniform dispersion of precipitates. 

Besides the effect on age strengthening, the prior-age stretch may also introduce cold work hardening and therefore endue increased strength to quench-stretched Al–Cu alloy sheets. This strengthening originates from the establishing of high density of dislocations that occurs during the stretching. It could depredate during the aging process due to the recovery when high aging temperature is applied, and the influence should be taken into account. 

To compare with the previous study, in which stretch up to 8% was applied, the draw-forming can induce strain up to 15% to the final components. The effect of pre-stretch can thus be quite different. It is necessary to examine the effect of a larger amount of prior-aging stretch, aiming to design post-forming aging process for thin-wall components of Al–Cu–x alloys. The current study thus aims to investigate the effect of pre-stretch on the precipitation and the mechanical performances of Al–Cu alloy. Two levels of temperature (150 and 170 °C) and three levels of pre-stretch up to 15% were selected to address the influences of the cold work and the subsequent aging in 2219 Al alloy. 

## 2. Experimental 

2219 Al alloys plate of T6 temper was selected as the starting material in the current study. The chemical composition of the alloy is shown in Table 1. The specimens used for the tensile test were cut off from the plate along the rolling direction of the plate and ground carefully before putting into the solution treatment. The cross-section of the samples was 6 × 1.5 mm^2^ and the gauge length was 18 mm. The specimens were solution treated in NaNO_3+_KNO_3_ molten salt for 20 min at 535 ± 3 °C before being quenched with cold water. Some specimens were stretched for 5%, 10% and 15% on an Instron 5569 tensile testing system at a strain rate of 2 mm/min before putting into aging. An extensometer was used to monitor the strain within the gauge, aiming to terminate the pre-tension timely. The quenched and quenched-stretched specimens were then aged at 150 and 170 °C for 2, 4, 8, 12, and 16 h, respectively. All these specimens were then put into the tensile testing on an tensile testing system (Instron 5569, Instron ITW, Norwood, MA, USA) at a strain rate of 2 mm/min, referring the standard of GB228-87. Three specimens were used for the tensile testing for each condition and the averaged data were used for evaluating the mechanical properties of specimens. The schematics of the sampling process are demonstrated in Figure 1.

Foils for transmission electron microscope (TEM, JEOL 2100, Tokyo, Japan) observation and differential scanning calorimetry (DSC) analysis were cut from the tensile specimens aged to various conditions. Foils selected for thermodynamics analysis were carefully polished to remove the surface layer before putting into the DSC system (Q2000, TA Instruments, New Castle, DE, USA) for analysis. Foils selected for TEM observation were mechanically polished to below 50 μm and then electro-thinned to fabricate the specimens. The microstructure was observed on a TEM and representative morphology and selected area electron diffraction (SAED) patterns were recorded for subsequent investigation. The strength of specimens of various conditions was measured on a Instron5569 tensile testing machine.

## 3. Results and Discussion

DSC was utilized to analyze the precipitation process of quench-stretched specimens. As a reference, DSC analysis was also performed on quench-aged specimens. The DSC curve of the specimen quenched and aged at 170 °C for 2 h presents two distinct endothermic peaks together with two exothermic peaks in 20~400 °C range, as shown in Figure 2a. The first endothermic peak detected at 204 °C is associated with the G.P. zones’ dissolution. This peak covers the 164~225 °C range, indicating a long-lasting dissolution. The second endothermic peak located at 241.4 °C is most likely attributed to the dissolution of *θ″* phases. The exothermic peak observed at 305.3 °C is attributed to the forming of *θ′* phases. The forming of *θ* phases contributes to the exothermic peak at 457.7 °C. The DSC curve is quite similar to that observed in Son [20] and Takeda [21], revealing the presences of G.P. zones, *θ″* phases and their transition to *θ′* and further *θ* phases. 

The DSC curves of quenched-stretched specimens are quite different, as shown in Figure 2a. The endothermic peak associated with G.P. zones’ dissolution disappears when the specimens are aged for more than 2 h, indicating the absence of G.P. zones. The endothermic peak associated with the dissolution of *θ″* phases is observed shift to lower temperature. Specifically, the peak shifts from 241.4 to 209.4 °C, 204.6 and 203.2 °C respectively as 5%, 10% and 15% stretch is applied. Also, the exothermic peak corresponding to *θ′* phases’ precipitation is observed shift to lower temperature as increased pre-stretch applied. This shift suggests that the dissolution of *θ″* and the precipitation of *θ′* phases occur at lower temperature to compare with the non-stretched specimens [22,23].

When aged for 4 h, only one endothermic peak and two exothermic peaks can be distinguished in the DSC curve of the non-stretched specimen, as shown in Figure 2b. The endothermic peak that located at 231 °C is attributed to the dissolution of *θ″* phases, meaning the presence of *θ″* phases and the absence of G.P. zones. When stretch is applied, the peak corresponding to the *θ″* phases’ dissolution shifts to lower temperature, similar to that observe in the case of 2 h. Meanwhile, the endotherm origins from *θ″* dissolution became weaker gradually as the stretch increases, indicating the decreased amount of remained *θ″* phases. The DSC curves of specimens aged for 8 h, as shown in Figure 2c are similar to those observed in the case of 4 h, but the endotherm/exotherm related to the precipitation process becomes much weaker. Specifically, the endothermic peak attributed to *θ″* phases’ dissolution becomes very low, indicating that the amount of *θ″* phases is quite small. For specimens aged for 16 h, only very weak endothermic/exothermic peaks can be distinguished on the DSC curves of stretched specimens, indicating the absence of G.P. zones and *θ″* phases, as shown in Figure 2d.

The DSC curves of specimens quenched, 10% stretched, and then aged at 150 °C for different times are demonstrated in Figure 3. The peaks associated with the dissolution of *θ″* phases and the forming of *θ′* does not shift apparently, which is different to that observed in case of 170 °C. These two peaks become weak gradually as the stretch increases but survive even after 16 h aging, indicating the remaining of *θ″* phases at the condition. This result reveals that the dissolution-precipitation is much slower at 150 °C and the influence of pre-stretch is partially moderated in comparation to those aged at 170 °C.

The morphology and the corresponding SAED of precipitates in specimens stretched for various amount and aged at 170 °C for 4 h are shown in Figure 4. High density of precipitates is observed shape up on the matrix in the non-stretched specimen. Most precipitates are 20~30 nm in length and 2~4 nm in thickness. The SAED pattern in Figure 4b indicates that most precipitates observed at this condition are *θ″* phases. The precipitation and coarsening accelerate apparently when 5% stretching is performed as substantive *θ′* phases are observed, as shown in Figure 4c,d. These *θ′* phases are 57 nm averagely in length and around 5 nm in thickness. A very fine *θ″* phase can be observed in intervals although the amount decreases, as demonstrated in the SAED pattern in Figure 4d. When the pre-stretch is further increased to 10%, *θ′* phase increases in number density but does not change apparently in size. The *θ′* phase in the 10% pre-stretched specimen was 58 nm averagely. Fine *θ″* phase can still be distinguished on the matrix, as demonstrated at the morphology in Figure 4e and the SAED in Figure 4f. When the pre-stretch is eventually increased to 15%, the morphology and the SAED patterns in Figure 4g,h revealed that the precipitates are mainly *θ′* phase along with a small amount of *θ″* phase. The amount of the *θ′* phases increase slightly in comparation with that in case of 10% stretching. The size of the *θ′* phases is measured as 57 nm, which does not change significantly with further increased amount of pre-stretching. 

The increase in *θ′* and the according decrease of *θ″* suggest that the pre-stretching facilitates the precipitation process. Specifically, it induces an accelerated development of *θ’* when aged at 170 °C. The high density of dislocations that developed during the stretching are believed to enhance the precipitation of *θ’* phases. In fact, the increase in dislocation density was proved in a previous research. The research [17] noted that the dislocation density in the as-quenched condition was low (~2 × 10^12^·m^−2^) but increases quickly by a factor of 100 or even higher when stretching was introduced. For instance, the dislocation density reaches at a high level of 1.8 × 10^14^·m^−2^ when a strain of 7% was introduced. The high density of dislocation facilitates *θ′* phases’ preferential separation from the matrix since *θ′* phases prefer to nucleate on dislocations to compare with *θ″*. Meanwhile, the high deformation stored energy accumulated during stretching helps to decrease the nucleation barriers [24]. On the other hand, the transition from existing *θ″* phases to *θ′* phases can also be enhanced since the high density of dislocation induce an increased diffusion. Additionally, the depletion of Cu due to the *θ″* phases’ preferential precipitation may further suppress the precipitation of *θ′* phases.

The precipitates in specimens stretched 10% and then aged for 16 h at 170 °C and 150 °C were observed on TEM, as shown in Figure 5. *θ′* phases are observed everywhere but *θ″* phases can hardly be observed in the specimen aged at 170 °C for 16 h, as shown in Figure 5a,b. This observation is consistent with the DSC analysis, revealing *θ″* phases’ fully transition to *θ′* phases. The *θ′* precipitates are 68 nm in length and 5–10 nm in thickness, a little larger than those in the specimen aged for 4 h. The morphology together with the SAED patterns demonstrated in Figure 5c,d reveals the coexisting of *θ″* and *θ′* phases when aged at 150 °C for 16 h, which is also consistent to the DSC analysis. The *θ′* phase is 56 nm averagely in size in the specimen aged at 150 °C, which is comparable to that in the specimen peak-aged at 170 °C. The comparison suggests that the precipitation kinetic is much slower at 150 °C than that at 170 °C and the transition from *θ″* to *θ′* cannot complete even when aged for a long time, which is similar to the previous study [25]. 

The size distribution of *θ′* phases in specimens of selected condition are shown in the histogram in Figure 6. The length distribution in specimens pre-stretched for 5%, 10% and 15% and aged for 4 h is similar to each other except that the fraction of *θ′* phases with length of 30~40 nm is little higher in specimen pre-stretched for 5%. When the aging proceeds to 16 h, *θ′* phases in specimen pre-stretched for 10% grow slightly, indicating a slowly developed over-aging in the alloy. It can also be deduced that the growth of *θ′* phases is quite slight when the aging duration is prolonged from 4 h to 8 h. 

The mechanical performances of specimens as-quenched and quenched-stretched are demonstrated in Figure 7. The as-quenched specimen presents low yield strength (YS) of 143 MPa but high elongation of 25.4%, presenting an excellent drawing formability. The YS is found increases persistently but the elongation decreases accordingly as increased pre-stretch is performed. YS rises to 265 MPa when 5% stretch is applied, indicating an increased deformation resistance. YS of 322 MPa is eventually obtained when the as-quenched specimens is pre-stretched for 15%, which is apparently higher than that of non-stretched specimens. The increase in the strength is evident when 5% stretched is applied and then mild off when the amount of pre-stretching further increases, which is favorable for draw forming. The tensile strength (TS) increases much more slightly comparing with the yield strength. The increase in the YS together with the decrease in the elongation is dominantly attributed to the increased dislocation density. It is indicated that the contribution of existing dislocations to the TS is much less to compare with that of the YS.

The variation in mechanical performances of as-quenched and quench-stretched specimens during the aging is shown in Figure 8. The YS of the non-stretched specimen presents a progressive response to the aging and reaches at the peak YS of 300 MPa after aged for 12 h. Also, the characteristic of over-aging develops quite slowly after the aging exceeds further. A much quicker increase of YS is observed at the first 4 h in all the pre-stretched specimens despite the diversity in the pre-stretching. The YS of quench-stretched specimen reached at the peak level within 4 h, which is much earlier than that in the as-quenched specimens and peak YS of 349, 372, and 390 MPa, which is much higher than the peak YS obtained in non-stretched specimens, was observed. A decrease in YS however develops immediately after reaching the peak-aged condition, as shown in the Figure 8a, indicating the accelerated developing towards over-aging condition. The diversity of YS between specimens stretched for different amount narrows away quickly after the aging time exceeds 4 h, and eventually disappears after 8 h, as shown in Figure 8a. Meanwhile the stretched specimens’ advantage over the non-stretched ones in YS decreases gradually from 70~100 MPa to around 35 MPa during the 4~8 h. The loss of age-hardening is partially responsible for the decreased YS since the *θ′* precipitates are found to coarsen quite slightly in the over-aged specimen. Moreover, the rapid decrease in YS is believed attributed to the recovery in the alloy during the aging since remarkable recovery can occur in cold-rolled 2219 sheet when aged at 175 °C [3]. High density of dislocations that developed during the stretching annihilates quickly and their contribution to the YS then reduces. 

The tensile strength (TS) of the specimens increases quickly within the first 4 h and reach at the level of 435 MPa or 460 MPa for the non-stretched specimens, as shown in Figure 8b, as the aging proceeds further. The stretched specimens present slightly higher TS comparing to the non-stretched specimens, but the amount of deformation does not influence the TS. The decrease of TS due to the over-aging, however, does not occur evidently when the aging proceeds further from the moment of 4 h, which is quite different from that observed in the YS. The comparative observing suggests that the TS is not significantly influenced by the limited coarsening of *θ′* precipitates. Moreover, the occurrence of recovery during aging is considered to have little influence on the TS. It is believed that the contribution of existing dislocations to the TS is much less to compare with that to the YS. The annihilation of dislocations, that developed in the later stage of the aging, then contributes much more to the decrease of YS than to the decrease of TS. In addition, the specimens exhibit comparable peak TS despite the different pre-stretching level. This can be adequately explained by the similar-sized *θ′* phase in the peak-aged specimens that observed in Figure 4. 

For all specimens, the elongation decreases as aging proceeds, as shown in Figure 8c. For each aging condition, the non-stretched specimens present much higher elongation to compare with the pre-stretched ones. For instance, the as-quenched specimen presents an elongation exceeding 26%, revealing a high formability, while the elongation in non-stretched specimens decreases to 18.2%, 14.3%, and 10.1% respectively. The decrease in the elongation is find near to that of the value of the pre-stretching for specimens of solution-treated condition. The decrease of elongation with the increases of pre-stretching was also observed in specimens aged to same conditions despite the varied pre-stretching. Specimens of peak-aged condition exhibits elongation of 14.4%, 10.7%, and 7.1%, respectively, when pre-stretching of 5%, 10%, and 15% were applied. 

The net increase in yield strength (NIY) during the aging is calculated and the results are demonstrated in Figure 8d. The NIYs are generally lower in the quenched-stretched specimens than that in the non-stretched ones. More specifically, the NIY decreases as the stretching decreases. The softening related to the recovery is enhanced when lager amount of stretch is applied, which then lead to decreased NIYs.

The influence of pre-stretching on the age-hardening was also examined at 150 °C introducing a 10% of stretch. As shown in Figure 9a, the age-hardening develops quickly during the first 2 h and subsequently slows down. The increase of strength continues all thorough the aging with a YS of 387 MPa obtained at 16 h. The elongation of stretched specimens increases slightly at the beginning of aging and reaches at a peak level of 15.9%. It then decreases as the aging proceeds and eventually to 11.2% after aged for 16 h when peak strength obtained. The YS of stretched specimens is around 100 MPa higher than that of the non-stretched ones all through the aging process, as shown in Figure 9b, which is higher and more stable than that in case of 170 °C. The disparity between the TS is around 50 MPa all through the aging, narrower than that in YS, as shown in the Figure 9b. The recovery is believed much weak due to the lower aging temperature, which thus leads to milder decreases in strength. 

Either aged at 150 °C or 170 °C, the peak-aged condition is associated with the co-exist of small amount of *θ″* and fine *θ′* phases, indicating fine *θ′* phases contribute much more to the strength to compare with the *θ″* or *θ* phases, which is consistent with the observation in previous research [20,26].

The current research presents both the intensified aging-hardening effect and the recovery induced softening that related to the prior-aging stretching within a wide range (~15%) of strain, which then extends the understanding of the stretch-aging to include the forming induced strain in. The positive response from the pre-stretching facilitates the fabricating of thin wall components along a newly proposed route of solution→draw forming→aging through which the precise forming and the aging-strength of Al–Cu alloy can be accomplished synergistically. Moreover, intensified precipitation, together with the enhanced aging-strengthening, due to the pre-stretching, has also been noted in the Ag-bearing Al–Cu–Mg alloys. Some research observed that the introducing of pre-stretching contribute to refining of Ω-phase and uniform dispersion of *θ′*-phase [14]. Pre-stretching is believed to disrupt the clustering of Ag–Mg with Cu atoms, which blocks the forming of GP zones and thereafter facilitates the precipitation of the *θ′*-phase [16]. The effect of stretching is similar in Ag-free and Ag-bearing Al–Cu–Mg alloy, which then extend the scope of the “solution→draw forming→aging” route. 

## 4. Conclusions

(1)The as-quenched 2219 alloy presents low YS (143 MPa) together with a high elongation (25.4%) during the tensile test, inferring an excellent formability in draw-forming processes.(2)The introducing of pre-stretch endues increased YS and thus can be helpful to exert the potential of the alloy. YS of 387.5 and 376.8 MPa is obtained when specimens pre-stretched for 10% is aged at 150 and 170 °C respectively, much higher than that of the peaked YS observed in the non-stretched specimens (319.2 MPa).(3)The precipitation of G.P. zones and the transition to *θ″* shifts to lower temperature when as-quenched Al–Cu alloy sheet is stretched for more than 5%. The high density of dislocations developed during the stretching contributed to the acceleration in precipitation. Precipitates in stretched specimens are mainly *θ′*-phase while mixed precipitates (mainly *θ″* and minor *θ′*) developed in non-stretched specimens.(4)Quench-stretched specimens present a much quicker age-hardening at the beginning stage, which endue higher peak YS. The YS, however, decrease much more quickly due to the recovery that occurs during the aging processes.(5)The current study thus indicates the feasibility of draw-forming thin-wall component of Al–Cu alloys and to obtained high strength via post-forming aging.

## Figures and Tables

**Figure 1 materials-14-02101-f001:**
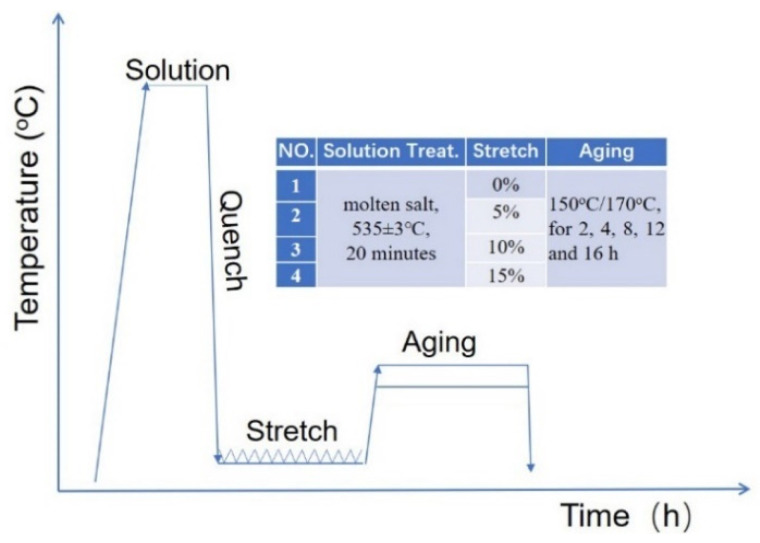
Schematics of the sampling plot in the current research.

**Figure 2 materials-14-02101-f002:**
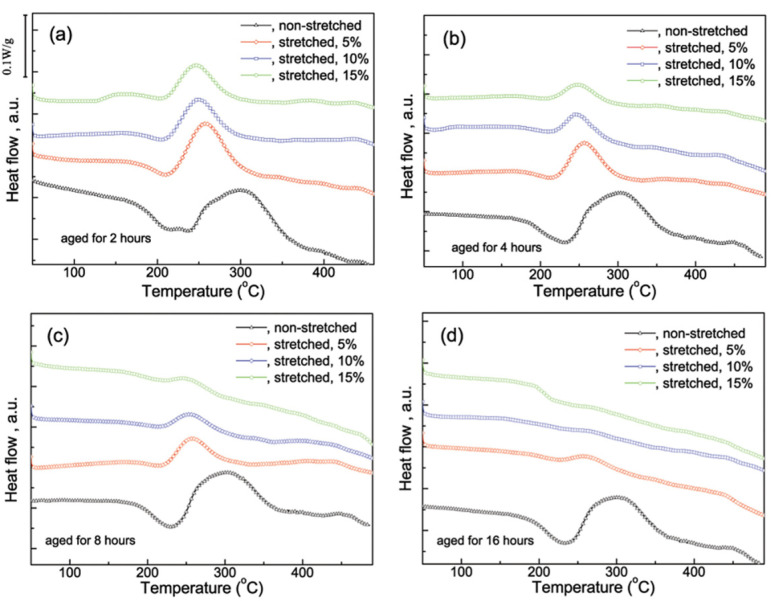
The DSC curves of as-quenched and quench-stretched specimens after aged at 170 °C for (**a**) 2 h, (**b**) 4 h, (**c**) 8 h and (**d**) 16 h.

**Figure 3 materials-14-02101-f003:**
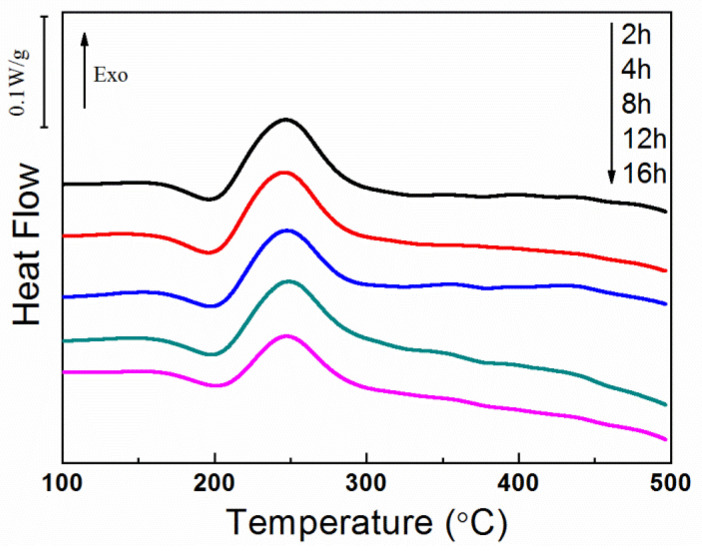
The differential scanning calorimetry (DSC) curves of as-quenched and 10% stretched specimens after aged at 150 °C.

**Figure 4 materials-14-02101-f004:**
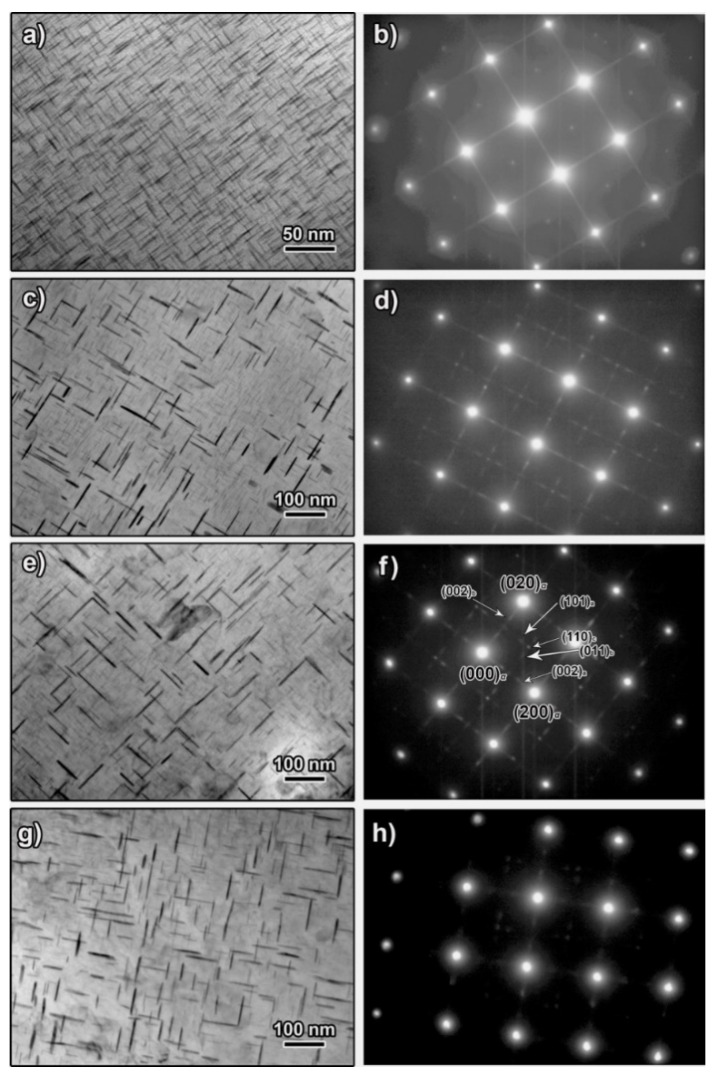
TEM morphology and corresponding selected area electron diffraction (SAED) patterns demonstrating the microstructure of specimens aged at 170 °C for 4 h. The specimens were non-stretched, (**a**,**b**), and then stretched for 5%, (**c**,**d**), 10%, (**e**,**f**) and 15%, (**g**,**h**).

**Figure 5 materials-14-02101-f005:**
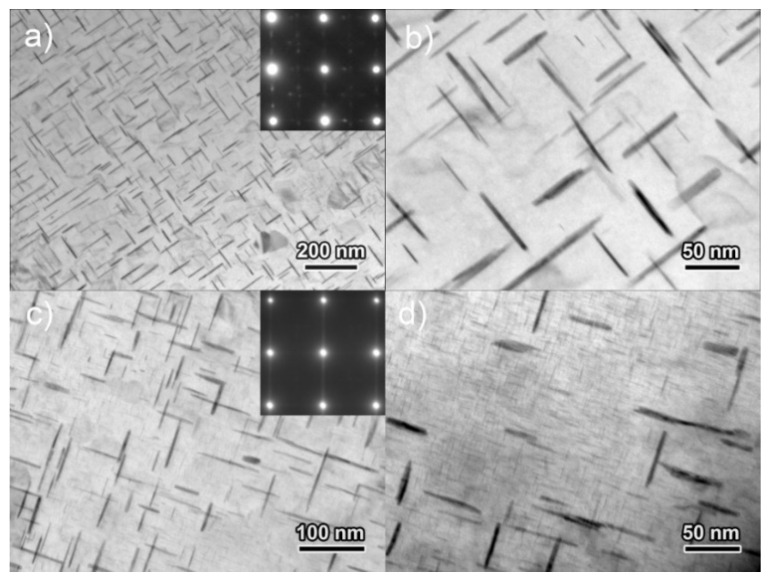
TEM images of the precipitates in the specimens that solution treated, stretched for 10% and then aged for 16 h at 170 °C, (**a**,**b**); or at 150 °C, (**c**,**d**).

**Figure 6 materials-14-02101-f006:**
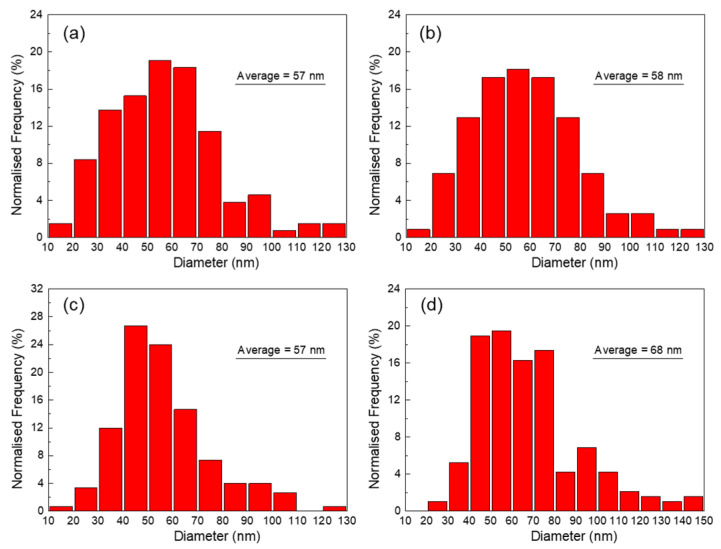
Size distribution of the precipitates in the specimens that processed by various ways. (**a**) Pre-stretched for 5% and aged at 170 °C for 4 h; (**b**) Pre-stretched for 10% and aged at 170 °C for 4 h; (**c**) Pre-stretched for 15% and aged at 170 °C for 4 h; (**d**) Pre-stretched for 10% and aged at 170 °C for 16 h.

**Figure 7 materials-14-02101-f007:**
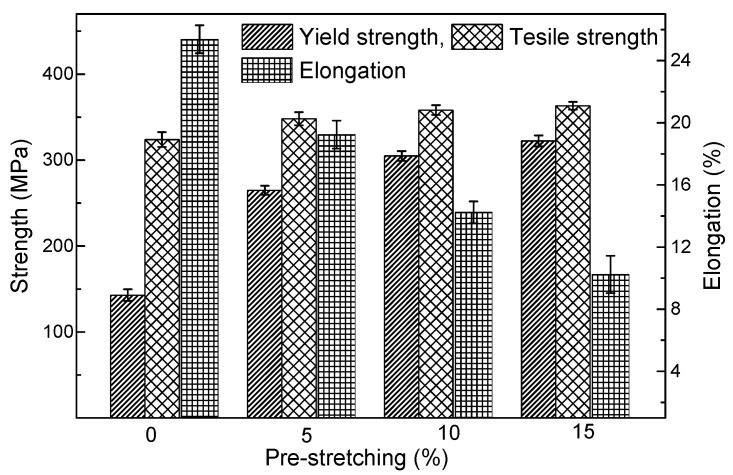
Mechanical performances of 2219 specimens that solution-treated and pre-stretched by various amount.

**Figure 8 materials-14-02101-f008:**
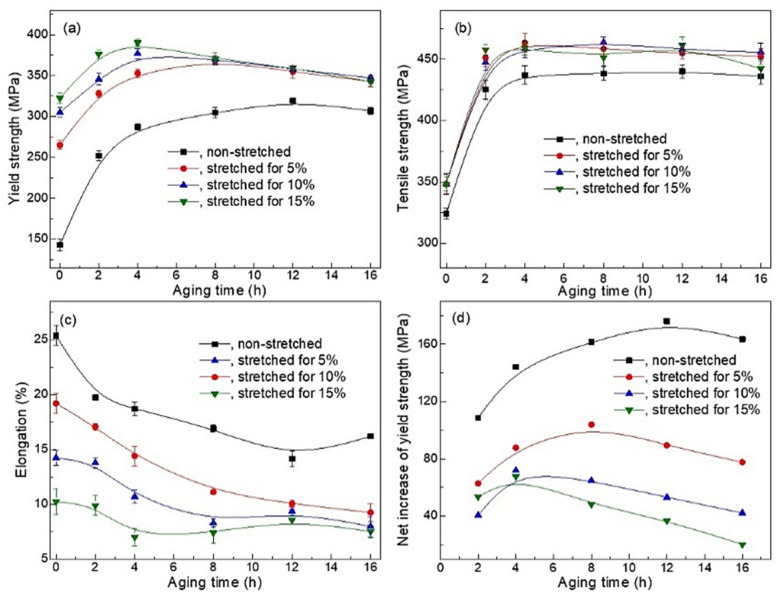
The performances of stretch-aged specimens, (**a**) yield strength, (**b**) tensile strength, (**c**) the elongation and (**d**), the net increase in yield strength. Specimens were stretched for 5%, 10% and 15% and before aged at 170 °C.

**Figure 9 materials-14-02101-f009:**
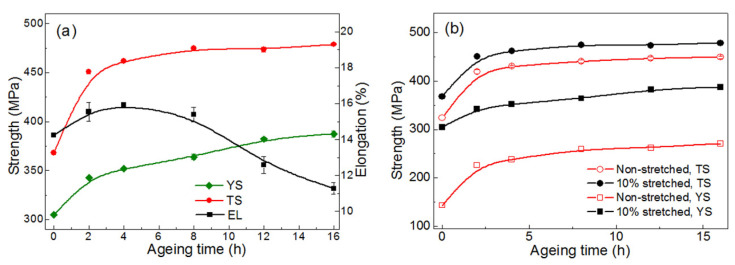
Mechanical properties of specimen as-quenched and stretch-aged when aged at 150 °C, (**a**), and the comparing to non-stretched one’s, (**b**).

**Table 1 materials-14-02101-t001:** Chemical composition (wt%) of the 2219 aluminum alloy.

Cu	Mn	V	Zr	Fe	Si	Al
6.12	0.33	0.10	0.13	0.13	0.046	Bal.

## Data Availability

The data presented in this study are available on request from the corresponding author.

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
