# Peer review of "Effect of Pre-Stretch on the Precipitation Behavior and the Mechanical Properties of 2219 Al Alloy"

_materials, 2021, doi:10.3390/ma14092101_

Round 1

Reviewer 1 Report

I am missing a discussion on the influence of stretch on size and size distribution of the strengthening phase and how those differences influence and affect strength. 

Also the comparison is not quite fair. The authors correctly state that stretch increases the aging kinetics. Kinetic differences result in mainly theta" strengthening phase in the unstretched material whereas in the stretched material the main strengthening phase is theta'. Theta" is not as potent of a strengthening phase as theta'. How large would the strength be in the non stretched condition, if it were aged up to theta'? 

It actually looks as if the size distribution of theta" in the unstretched condition is finer and more uniform. Should it not be shown what the unstretched condition size distribution looks like when aged to theta'. The absence of dislocations and nucleation sites may result in a larger coarser theta' phase and therefore lower properties, but that should be proven out. 

It should also be shown that indeed recovery is what leads to lowering in TYS with extended aging and not simply overaging due to the enhanced kinetics. The reduction in elongation and strength could be due to accelerated coarsening (Ostwald ripening) of theta' phase on dislocation cores thereby reducing the strengthening phase in the matrix rather than recovery.

It would also be good to discuss the actual composition of the alloy. Discuss a little bit the difference in stretch importance to Ag bearing 2xxx Al-Cu-Mg alloys.

Author Response

Reply to the reviewers’ comments-1

Thanks a lot for the reviewers’ professional comments on the manuscript that we submitted to the journal of Materials (No. Materials_1136699). The authors have revised the manuscript carefully regarding on the comments.

Comments 1-1 I am missing a discussion on the influence of stretch on size and size distribution of the strengthening phase and how those differences influence and affect strength. 

Reply to comment 1-1

The size of the strengthening phase θ′ phase and the related influence on strength are involved in the revised manuscript. The specimens pre-stretched by different amount have similar-sized θ′ phase when they are aged at 170 oC to peak condition. Meanwhile, the tensile strength is found to be comparable in the specimens pre-stretched by different amount and peak-aged. Therefore, the difference in the strengthening from θ′ phase is considered insignificant.

The size distribution was demonstrated in the revised manuscript by adding an figure (Fgiure.6) and the related description.

The corresponding revisions are as follows—

“The precipitation and coarsening accelerate apparently when 5% stretching is performed as substantive θ′ phases are observed, as shown in Fig.4c and 4d. These θ′ phases are 57 nm averagely in length and around 5 nm in thickness. Very fine θ″ phase can be observed in intervals although the amount decreases, as demonstrated in the SAED pattern in Fig.4d. When the pre-stretch is further increased to 10%, θ′ phase increases in number density but does not change apparently in size. The θ′ phase in the 10% pre-stretched specimen was 58 nm averagely. Fine θ″ phase can still be distinguished on the matrix, as demonstrated at the morphology in Fig.4e and the SAED in Fig.4f. When the pre-stretch is eventually increased to 15%, the morphology and the SAED patterns in Fig.4g and Fig.4h revealed that the precipitates are mainly θ′ phase along with a small amount of θ″ phase. The amount of the θ′ phases increase slightly in comparation with that in case of 10% stretching. The size of the θ′ phases is measured as 57 nm, which does not change significantly with further increased amount of pre-stretching.”

Figure 4. TEM morphology and corresponding SAED patterns demonstrating the microstructure of specimens aged at 170°C for 4 hours. The specimens were non- stretched, (a) and (b), and then stretched for 5%, (c) and (d), 10%, (e) and (f) and 15%, (g) and (h).

“The specimens exhibit comparable peak TS despite the different pre-stretching level. This can be adequately explained by the similar-sized θ′ phase in the peak-aged specimens that observed in Fig.4.” (Page 7 in the revised manuscript)

The size distribution of θ′ phases in specimens of selected condition are demonstrated in the histogram in Fig.6. The length distribution in specimens pre-stretched for 5%, 10% and 15% and aged for 4 hours is similar to each other except that the fraction of θ′ phases with length of 30~40 nm is little higher in specimen pre-stretched for 5%. When the aging proceeds to 16 h, θ′ phases in specimen pre-stretched for 10% grow slightly, indicating a slowly developed over-ageing in the alloy. It can also be deduced that the growth of θ′ phases is quite slight when the ageing duration is prolonged from 4 hours to 8 hours.     

Figure 6. Size distribution of the precipitates in the specimens that processed by various ways. (a) Pre-stretched for 5% and aged at 170 oC for 4h; (b) Pre-stretched for 10% and aged at 170 oC for 4h; (c) Pre-stretched for 15% and aged at 170 oC for 4h; (d) Pre-stretched for 10% and aged at 170 oC for 16h.

Comments 1-2 Also the comparison is not quite fair. The authors correctly state that stretch increases the aging kinetics. Kinetic differences result in mainly theta" strengthening phase in the unstretched material whereas in the stretched material the main strengthening phase is theta'. Theta" is not as potent of a strengthening phase as theta'. How large would the strength be in the non-stretched condition, if it were aged up to theta'? 

Reply to comment 1-2

The research focused basically on the draw-forming and the substantial aging-strengthening. Specimens of peak-aged condition, received varied pre-stretch before ageing, was used to evaluate the strength of the formed components.      

On the other hand, precipitates of theta" cannot be transformed completely to theta’ in the non-stretched specimens, according to the observing either in the current or in the previous research [R1-R5]. After aged at 170oC for 12h, the non-stretched specimen exhibits the peak strength which is lower than that of pre-stretched specimen. The higher strength in the pre-stretched specimens is partially attributed to the accelerated transformation of theta" phase to theta' phase and partially to the introduced dislocation strengthening.

Figure 8. the performances of stretch-aged specimens, (a) yield strength, (b) tensile strength, (c) the elongation and (d), the net increase in yield strength. Specimens were stretched for 5%, 10% and 15% and before aged at 170 °C.

[R1]  Lu, Y.; Wang, J.; Li, X.; Chen, Y.; Zhou, D.; Zhou, G.; Xu, W. Effect of pre-deformation on the microstructures and properties of 2219 aluminum alloy during aging treatment. J. Alloys Compd. 2017, 699, 1140-1145, doi:https://doi.org/10.1016/j.jallcom.2016.12.006.

[R2]  Wang, H.; Yi, Y.; Huang, S. Influence of pre-deformation and subsequent ageing on the hardening behavior and microstructure of 2219 aluminum alloy forgings. J. Alloys Compd. 2016, 685, 941-948, doi:https://doi.org/10.1016/j.jallcom.2016.06.111.

[R3]  Li, Z.X.; Zhan, M.; Fan, X.G.; Wang, X.X.; Ma, F. Age hardening behaviors of spun 2219 aluminum alloy component. Journal of Materials Research and Technology 2020, 9, 4706-4716, doi:https://doi.org/10.1016/j.jmrt.2020.02.098.

[R4]  Liu, H.; Bellón, B.; Llorca, J. Multiscale modelling of the morphology and spatial distribution of θ′ precipitates in Al-Cu alloys. Acta Mater. 2017, 132, 611-626, doi:https://doi.org/10.1016/j.actamat.2017.04.042.

[R5]  Ma, P.P.; Liu, C.H.; Wu, C.L.; Liu, L.M.; Chen, J.H. Mechanical properties enhanced by deformation-modified precipitation of θ′-phase approximants in an Al-Cu alloy. Mater. Sci. Eng. A 2016, 676, 138-145, doi:https://doi.org/10.1016/j.msea.2016.08.068.

[R6]  Ringer, S.P.; Muddle, B.C.; Polmear, I.J. Effects of cold work on precipitation in Al-Cu-Mg-(Ag) and Al-Cu-Li-(Mg-Ag) alloys. Metall. Mater. Trans. A 1995, 26, 1659-1671, doi:10.1007/BF02670753.

[R7]  Gazizov, M.; Kaibyshev, R. Effect of pre-straining on the aging behavior and mechanical properties of an Al–Cu–Mg–Ag alloy. Mater. Sci. Eng. A 2015, 625, 119-130, doi:https://doi.org/10.1016/j.msea.2014.11.094.

Comments 1-3 It actually looks as if the size distribution of theta" in the unstretched condition is finer and more uniform. Should it not be shown what the unstretched condition size distribution looks like when aged to theta'. The absence of dislocations and nucleation sites may result in a larger coarser theta' phase and therefore lower properties, but that should be proven out. 

Reply to comment 1-3

The size distribution of theta" phase in the unstretched condition looks like finer and more uniform than theta" phase in the stretched specimen when these specimens are aged at 170oC for 4h, as shown in Fig. 4. This is because that the theta" phase has largely transformed to theta' phase in the stretched specimen. The DSC curves suggests that the transformation of theta" phase to theta' phase is lowered in the unstretched specimen than that in the stretched specimen, which is consistent with the previous studies [R1-R5]. Therefore, it is reasonable that the strengthening contribution from theta' phase is smaller in the unstretched specimen relative to that in the stretched specimen. The results of tensile properties in the present work exactly accord with this law. Moreover, as the reply to comment 1-4, the dislocation strengthening introduced by the pre-stretching also contributes to the higher properties in the pre-stretched specimen.

Figure 4. TEM morphology and corresponding SAED patterns demonstrating the microstructure of specimens aged at 170°C for 4 hours. The specimens were non-stretched, (a) and (b), and then stretched for 5%, (c) and (d), 10%, (e) and (f) and 15%, (g) and (h).

[R1]  Microstructures and properties of 2219 aluminum alloy during aging treatment. J. Alloys Compd. 2017, 699, 1140-1145, doi:https://doi.org/10.1016/j.jallcom.2016.12.006.

[R2]  Wang, H.; Yi, Y.; Huang, S. Influence of pre-deformation and subsequent ageing on the hardening behavior and microstructure of 2219 aluminum alloy forgings. J. Alloys Compd. 2016, 685, 941-948, doi:https://doi.org/10.1016/j.jallcom.2016.06.111.

[R3]  Li, Z.X.; Zhan, M.; Fan, X.G.; Wang, X.X.; Ma, F. Age hardening behaviors of spun 2219 aluminum alloy component. Journal of Materials Research and Technology 2020, 9, 4706-4716, doi:https://doi.org/10.1016/j.jmrt.2020.02.098.

[R4]  Liu, H.; Bellón, B.; Llorca, J. Multiscale modelling of the morphology and spatial distribution of θ′ precipitates in Al-Cu alloys. Acta Mater. 2017, 132, 611-626, doi:https://doi.org/10.1016/j.actamat.2017.04.042.

[R5]  Ma, P.P.; Liu, C.H.; Wu, C.L.; Liu, L.M.; Chen, J.H. Mechanical properties enhanced by deformation-modified precipitation of θ′-phase approximants in an Al-Cu alloy. Mater. Sci. Eng. A 2016, 676, 138-145, doi:https://doi.org/10.1016/j.msea.2016.08.068.

Comments 1-4 It should also be shown that indeed recovery is what leads to lowering in TYS with extended aging and not simply overaging due to the enhanced kinetics. The reduction in elongation and strength could be due to accelerated coarsening (Ostwald ripening) of theta' phase on dislocation cores thereby reducing the strengthening phase in the matrix rather than recovery.

Reply to comment 1-4

The corresponding revisions are as follows—

[R3]  Li, Z.X.; Zhan, M.; Fan, X.G.; Wang, X.X.; Ma, F. Age hardening behaviors of spun 2219 aluminum alloy component. Journal of Materials Research and Technology 2020, 9, 4706-4716, doi:https://doi.org/10.1016/j.jmrt.2020.02.098.

Comments 1-5  It would also be good to discuss the actual composition of the alloy. Discuss a little bit the difference in stretch importance to Ag bearing 2xxx Al-Cu-Mg alloys.

Reply to comment 1-5

The actual composition of the 2219 alloy applied in the present work has been added in the revised manuscript, as shown in Table.R1. The alloy has a nominal composition of Al-6.12Cu-0.33Mn (wt%) and free of Ag. The present work is focused on the effect of pre-stretching on the precipitation and the mechanical performances of Al-Cu alloy, aiming to design post-forming ageing process for thin-wall components of Al-Cu-x alloys. Intensified precipitation, together with the enhanced aging-strengthening, due to the pre-stretching, has also been noted in the Ag-bearing Al-Cu-Mg alloys. Some research observed that the introducing of pre-stretching contribute to refining of Ω-phase and uniform dispersion of θ′-phase [R6]. Pre-stretching is believed to disrupt the clustering of Ag–Mg with Cu atoms, which blocks the forming of GP zones and thereafter facilitates the precipitation of the θ′-phase [R7]. The effect of stretching is similar in Ag-free and Ag-bearing Al-Cu-Mg alloy, which then extend the scope of the “draw forming-ageing” route.  

Table R1. Chemical composition (wt %) of the 2219 aluminum alloy.

Cu

Mn

V

Zr

Fe

Si

Al

6.12

0.33

0.10

0.13

0.13

0.046

Bal.

The corresponding revisions are as follows—

Intensified precipitation, together with the enhanced aging-strengthening, due to the pre-stretching, has also been noted in the Ag-bearing Al-Cu-Mg alloys. Some research observed that the introducing of pre-stretching contribute to refining of Ω-phase and uniform dispersion of θ′-phase [R6]. Pre-stretching is believed to disrupt the clustering of Ag–Mg with Cu atoms, which blocks the forming of GP zones and thereafter facilitates the precipitation of the θ′-phase [R7]. The effect of stretching is similar in Ag-free and Ag-bearing Al-Cu-Mg alloy, which then extend the scope of the “draw forming-ageing” route. 

[R6]      Ringer, S.P.; Muddle, B.C.; Polmear, I.J. Effects of cold work on precipitation in Al-Cu-Mg-(Ag) and Al-Cu-Li-(Mg-Ag) alloys. Metall. Mater. Trans. A 1995, 26, 1659-1671, doi:10.1007/BF02670753.

[R7]      Gazizov, M.; Kaibyshev, R. Effect of pre-straining on the aging behavior and mechanical properties of an Al–Cu–Mg–Ag alloy. Mater. Sci. Eng. A 2015, 625, 119-130, doi:https://doi.org/10.1016/j.msea.2014.11.094.

Reviewer 2 Report

REVIEW REPORT – materials_1136699

Article

Effect of pre-stretch on the precipitation behaviours and the mechanical properties of 2219 Al alloy

Authors

Guo-Ai Li, Zheng Ma, Jian-Tang Jiang, Wen-Zhu Shao, Wei Liu, and Liang Zhen

Mechanical stretch and heat treatment were applied aiming to design the manufacturing technology for thin wall components. Pre-stretching operation and solution ageing treatment have been used to achieve age-strengthening of 2219 Al alloy for draw-formed sheets.

Even if the study is based on well-known mechanical/thermal treatment methods for Al alloy, advanced imaging investigation (TEM – JEOL 2100), thermodynamics analysis (DSC) and SAED patterns have been used to examine the effect of mechanical/thermal treatment on the precipitation behaviour and morphology of the specimens.

The manuscript is well structured, and the discussion section cover the main aspects regarding the material structure, the influence of the treatments applied and the improvement of its properties. Conclusions are drawn, according to the results obtained and refer to the quantitative and qualitative aspects.

However, I would recommend a detailed description of the pre-stretching operation which has been carried out on Instron 5569 machine. Also, it would be needed to depict the samples, the tensile test schema, and parameters, and how the elongation has been measured, eventually the standard followed.

I noticed that references section must be improved with more recent articles, and I listed below some works which may be considered for the state-of-the art.

Chen, Y., Zhang, Z., Tsalanidis, A., Weyland, M., Medhekar, N.V., Bourgeois, L. The enhanced theta-prime (θ′) precipitation in an Al-Cu alloy with trace Au additions (2017) Acta Materialia, 125, pp. 340-350. Gorbatov, O.I., Gornostyrev, Y.N., Korzhavyi, P.A. Many-body mechanism of Guinier-Preston zones stabilization in Al–Cu alloys (2017) Scripta Materialia, 138, pp. 130-133. Liu, H., Bellón, B., LLorca, J. Multiscale modelling of the morphology and spatial distribution of θ′ precipitates in Al-Cu alloys (2017) Acta Materialia, 132, pp. 611-626.

Author Response

Reply to the reviewers’ comments-2

Thanks a lot for the reviewers’ professional comments on the manuscript that we submitted to the journal of Materials (No. Materials_1136699). The authors have revised the manuscript carefully regarding on the comments.

Comments 2-1 However, I would recommend a detailed description of the pre-stretching operation which has been carried out on Instron 5569 machine. Also, it would be needed to depict the samples, the tensile test schema, and parameters, and how the elongation has been measured, eventually the standard followed.

Reply to comment 2-1

The process of pre-stretching operation, including the depict of the samples, the tensile test schema and the parameters, together with the measuring of the elongation, has been described in detail in the revised manuscript. Also, the standard that followed was also presented.

The corresponding revisions are as follows—

Specimens used for tensile test were cut off from the plate along the rolling direction of the plate and ground carefully before putting into the solution treatment. The cross-section of the samples was 6×1.5 mm2 and the gauge length was set 18 mm.

Some specimens were stretched for 5%, 10% and 15% on an Instron 5569 tensile testing system at a strain rate of 2mm/min before putting into ageing, referring the standard of GB228-87. An extensometer was used to monitor the strain within the gauge, aiming to terminate the pre-tension timely.

All these specimens were then put into the tensile testing on an Instron 5569 tensile testing system at a strain rate of 2mm/min, referring the standard of GB228-87. Three specimens were used for the tensile testing for each condition and the averaged data were used for evaluating the mechanical properties of specimens.

Comments 2-2 I noticed that references section must be improved with more recent articles, and I listed below some works which may be considered for the state-of-the art.

Reply to comment 2-2

Some references have been added into the section of references in the revised manuscript.

[8] Chen, Y.; Zhang, Z.; Tsalanidis, A.; Weyland, M.; Medhekar, N.V.; Bourgeois, L. The enhanced theta-prime (θ′) precipitation in an Al-Cu alloy with trace Au additions. Acta Materialia, 2017, 125, 340-350.

[9] Gorbatov, O.I.; Gornostyrev, Y.N.; Korzhavyi, P.A. Many-body mechanism of Guinier-Preston zones stabilization in Al–Cu alloys. Scripta Materialia, 2017, 138, 130-133.

[10] Liu, H.; Bellón, B.; Llorca, J. Multiscale modelling of the morphology and spatial distribution of θ′ precipitates in Al-Cu alloys, Acta Materialia, 2017,132, 611-626.

[16] Gazizov, M.; Kaibyshev R., Effect of pre-straining on the aging behavior and mechanical properties of an Al–Cu–Mg–Ag alloy, Mater. Sci. Eng. A 2015 625, 119-130.

[17] Ma, P.P.; Liu, C.H.; Wu C.L.; Liu L.M.; Chen, J.H. Mechanical properties enhanced by deformation-modified precipitation of θ′-phase approximants in an Al-Cu alloy. Mater. Sci. Eng. A 2016, 676, 138–145.

Reviewer 3 Report

The article deals with analysis of effect of pre-stretch on the precipitation behaviour, determining the mechanical properties of control specimens and in case of pre-stretched samples and aged at different temperature. The information is presented in a logical way and the explanations and conclusions are good, but the novelty of this work should be highlighted. Major improvements (which are listed in attached file) are necessary to make the paper more understandable and proper for publication.

Author Response

Reply to the reviewers’ comments-3

Thanks a lot for the reviewers’ professional comments on the manuscript that we submitted to the journal of Materials (No. Materials_1136699). The authors have revised the manuscript carefully regarding on the comments. 

Comment 3-1 The notations that first appear in the text should be explained (eg DSC, TEM, T6, G.P., SAED, TS)

Reply to comment 3-1

The notations were explained when first appear in the revised manuscript.

The corresponding revisions are as follows—

The precipitation of Guinier-Preston zone (G.P. zones) and the transition to θ″ shifts to lower temperature when pre-stretched is performed.

Foils for transmission electron microscope (TEM, JEOL 2100) observing and differential scanning calorimetry (DSC) analysis were cut from the tensile specimens aged to various conditions.

…TEM and representative morphology and selected area electron diffraction (SAED) patterns were recorded for subsequent investigation.

Comments 3-2 The majority of the introduction contains a list of researches that have previously been published. The authors should consider improving this section by giving the results of these research (more specific data about similar studies on Al alloys) rather than given a general idea about these.

Reply to comment 3-2

The introduction of the manuscript has been checked and improved in the revised version. Some specific data about the research in the field of stretch-aging were added into the introduction. Also, a few references were added in to the section of references accordingly to reflect the research in the field. 

The corresponding revisions are as follows—

The precipitation sequence of Al-Cu alloys has been investigated extensively [6-10] and the effect of the stretch prior to ageing on the precipitation has been specifically concerned [11-15]. Small amounts (1~8%) of pre-stretch were observed to enhance the age hardening response in a few of Al-Cu-x alloys [13-15]. For instance, Ünlü [13] observed an enhanced age-hardening in an Al-Cu-Mg alloy both in natural and artificial ageing process when 6% of stretch was performed on the specimens of solution condition. Ringer [14] also observed similar effect in pre-stretched specimens at the early stage of natural ageing. It was also noted [16] that the yield strength of 2219 forgings increase by 26.2% when a 3% pre-deformation was induced before the ageing. The increase in strength was attributed to the combined effects of heterogeneous nucleation and growth of precipitates at dislocations due to the pre-deforming. The intensified aging-hardening is believed to root from the enhanced precipitation. It was noted that the introducing of pre-strain can stop the formation of GP zone and facilitate the precipitation of the θ'-phase and the S'-phase along dislocations [17]. The interaction between gliding dislocations and Cu atoms leads to the forming of the θ'-phase. Liu [20] introduced deformation of 2% to Al-Cu alloy and observed an increased age-hardening response, which was associated with the accelerated heterogeneous formation of θ' precipitates on dislocations. The presence of high density of dislocations that developed during the deformation is believed promote the ageing since they can serve as preferential matrix nucleation and growth sites for precipitates [21]. Ringer [14] observed that the application of a mechanical stretch enhanced the nucleation of θ’ phases in Al-Cu-Mg alloy, leading to a more uniform dispersion of precipitates.

Comment 3-3 The main originality and most important results are not indicated, please significantly improve this section.

Reply to comment 3-3

The main originality and most important results have been indicated in the end of the section of Results and discussion in the revised manuscript. 

The corresponding revisions are as follows—

The research presents both the intensified aging-hardening effect and the recovery induced softening that related to the prior-ageing stretching in a wide range (~15%) of strain, which then extends the understanding of the stretch-ageing evidently to include the forming induced strain in. The positive responds from pre-stretching facilitates the fabricating of thin wall components along a new route of solution→draw forming→ageing through which the precise forming and the aging-strength of Al-Cu alloy can be accomplished synergistically.

Comment 3-4 The sampling design – could you put a picture/schematics of the sampling plot? It would be interesting and easier to understand in a visual/graphical picture.

Reply to comment 3-4

A figure has been added into the manuscript to demonstrated the sampling process.

The Corresponding revisions are as follows—

The schematics of the sampling process was demonstrated in Fig.1.

Figure 1. Schematics of the sampling plot in the current research

Comment 3-5 How many samples from each type of treatment were investigated? What were the thermal regimes and the duration of exposure of the samples? It would be preferable to use a table with the types of samples and the treatments they have undergone.

Reply to comment 3-5

3 samples were used for each type of treatment and the thermal regimes and the duration of exposure of the samples were demonstrated via the sampling schematics (as shown in Fig.1). The table that the reviewer suggested has also been insert into the schematics.

The corresponding revisions regarding the description are as follows—

All these specimens were then put into the tensile testing on an Instron 5569 tensile testing system at a strain rate of 2mm/min, referring the standard of GB228-87. Three specimens were used for the tensile testing for each condition and the averaged data were used for evaluating the mechanical properties of specimens.

The schematics of the sampling process was demonstrated in Fig.1. in the revised manuscript.

Comment 3-6 Some specimens were stretched for 5%, 10% and 15% - How did you set these values and against which value are the percentages calculated?

Reply to comment 3-6

The stretching was set at 5%, 10% and 15% respectively considering the stretching that the sheet received during the draw-forming process. Quite a few processes were investigated to trace the distribution of the strain in the finished parts and a strain ranging from 4% to 14% was detected. A series of pre-stretching, 5%, 10% and 15%, was then carried out to reproduce and to exam the effect from the strain.

For the calculation of pre-stretching was based on the real-timed strain measured. Specifically, an extensometer was used to monitor the strain within the gauge aiming to terminate the pre-tension timely. According to the measuring process, the percentage of the pre-stretch was against the whole gauge length (18 mm in the current research).

The corresponding revisions are as follows—

The practice fabricating process of a fuselage indicated that plastic strain up to 15%, which is much larger than that applied in the conventional post-aging stretching, develops in the formed component during the draw-forming process. The large cold stretching applied during the forming, together with the following ageing, will then dominates mechanical performances of the finished components. To figure out the effect of pre-stretch in a wide range is then a basic mission prior to design fabrication process for draw-forming thin wall components.

Some specimens were stretched for 5%, 10% and 15% on an Instron 5569 tensile testing system at a strain rate of 2mm/min before put into ageing, referring the standard of GB228-87. An extensometer was used to monitor the strain within the gauge aiming to terminate the pre-tension timely.

Comment 3-7 In Figure 5, the representation of the values for the secondary axis of the graph is not very clear.

Reply to comment 3-7

The representation of the values for the secondary axis of the graph in Fig.5 was revised in the revised manuscript. Also, the figure caption of fig.5 was re-written. Also error bars have been added into the histogram in the revised manuscript.

The corresponding revisions are as follows—

Figure 6. Mechanical performances of specimens of 2219 alloys. These specimens were solution-treated and some of them were stretched to 5%, 10% and 15% respectively.

Comment 3-8 It would be interesting to introduce the characteristic stress-strain curves of the tested samples, including the control sample.

Reply to comment 3-8

The description of the stress-strain curves was carefully revised referring to the comments. 

The corresponding revisions are as follows—

The variation in mechanical performances of as-quenched and quench-stretched specimens during the ageing is shown in Fig.6. The yield strength (YS) of the stretch-free specimen presents a gradual response to the aging treatment and reaches at the peak YS of 300 MPa after aged for 12 hours. Also, the characteristic of over-aging develops quite slowly after the ageing exceeds further. A much quicker increase of YS is observed at the first 4 hours for all specimens despite the diversity in the pre-stretch amount. The YS of quench-stretched specimen reached at the peak level within 4 hours, which is much earlier than that in the as-quenched specimens. Peak YS of 349MPa, 372 MPa and 390 MPa was observed at specimen quenched and then stretched for 5%, 10% and 15% respectively after aged at 170oC for 4 hours, which is much higher than the peak YS obtained in stretch-free specimens. An apparent decrease in YS however occurs immediately after reaching the peak-aged condition, as shown in the Fig.6a, indicating the accelerated developing towards over-aging condition. The diversity of YS between specimens stretched for different amount narrows away quickly after the ageing time exceeds 4 hours, and eventually disappears after 8 hours, as shown in Fig.6a. Meanwhile the stretched specimens’ advantage over the non-stretched ones in YS decreases gradually from 70~100 MPa to around 35 MPa during the 4~8 hours. The rapid decrease in YS is attributed to the recovery in the alloy during the ageing. High density of dislocations that developed during the stretching annihilates quickly and their contribution to the YS then reduces.

The tensile strength (TS) of the specimens increases quickly within the first 4 hours and reach at the level of 435MPa or 460 MPa for the stretched-free quench-stretched specimens, as shown in Fig.6b, as the aging proceeds further. The stretched specimens present slightly higher TS comparing to the as-quenched specimens but the amount of deformation does not influence the TS apparently, which is different from that observed in case of YS. The decrease of TS due to the over-ageing, however, does not occur evidently when the ageing proceeds further from the moment of 4 hours, which is quite different from that observed in the YS. The comparative observing suggests that the occurrence of recovery does not influence the TS. It is believed that the contribution of existing dislocations to the TS is much less to compare with that to the YS. The annihilation of dislocations, that developed in the later stage of the aging, then contributes much more to the decrease of YS that to that of TS.

Comment 3-9 Due to the fact that the samples were not established according to the applied treatments, in the stage of describing the materials, the chapter Results seems ambiguous and difficult to understand. Maybe some correlation graphs should be made between the analyzed parameters.

Reply to comment 3-9

The describing of the materials and the results have been revised.

Reviewer 4 Report

This article discusses the effect of prestretching on the mechanical properties of 2219 alloy sheets in order to improve the strength characteristics. Since the alloys of this system are still important structural elements, improving their properties is an urgent task. The article is well-structured, well-structured and presents interesting results, confirmed by modern research methods.
However, there are a number of comments on this work that do not diminish the overall merits of the authors.
1) Please check the section numbering in the article
2) Is it necessary to add information about the samples on which the mechanical properties tests were carried out in the research methods? How many samples have been tested?
3) "A quick increase of YS is observed at the first 4 hours for all specimens despite the diversity in the pre-stretch amount." additional clarification required
4) How many samples have been tested? What is the confidence interval for the values ​​obtained when testing the specimens in Figure 5?
5) What is the reproducibility of the results? Or were the tests carried out on one sample? Confidence intervals should be plotted in Figures 6 a-c, as well as information on Fig. 6d should be entered in the figure caption.

Author Response

Reply to the reviewers’ comments-4

Thanks a lot for the reviewers’ professional comments on the manuscript that we submitted to the journal of Materials (No. Materials_1136699). The authors have revised the manuscript carefully regarding on the comments. 

Comment 4-1 Please check the section numbering in the article 

Reply to comment 4-1

The section numbering has been checked and revised.

Comment 4-2 Is it necessary to add information about the samples on which the mechanical properties tests were carried out in the research methods? How many samples have been tested?

Reply to comment 4-2

The information about the samples and the test, together with the number of samples, was added into the manuscript in the revised version.

The corresponding revisions are as follows—

Specimens used for tensile test were cut off from the plate along the rolling direction of the plate and ground carefully before putting into the solution treatment. The cross-section of the samples was 6×1.5 mm2 and the gauge length was set 18 mm.

Some specimens were stretched for 5%, 10% and 15% on an Instron 5569 tensile testing system at a strain rate of 2mm/min before put into ageing, referring the standard of GB228-87. An extensometer was used to monitor the strain within the gauge aiming to terminate the pre-tension timely.

All these specimens were then put into the tensile testing on an Instron 5569 tensile testing system at a strain rate of 2mm/min, referring the standard of GB228-87. Three specimens were used for the tensile testing for each condition and the averaged data were used for evaluating the mechanical properties of specimens.

Comment 4-3 "A quick increase of YS is observed at the first 4 hours for all specimens despite the diversity in the pre-stretch amount."/ additional clarification required

Reply to comment 4-3

The description was re-written by adding clarified information into the revised manuscript.

The corresponding revisions are as follows—

The tensile strength (TS) of the specimens increases quickly within the first 4 hours and reach at the level of 435MPa or 460 MPa for the stretched-free quench-stretched specimens, as shown in Fig.6b, as the aging proceeds further. The stretched specimens present slightly higher TS comparing to the as-quenched specimens but the amount of deformation does not influence the TS apparently, which is different from that observed in case of YS. The decrease of TS due to the over-ageing, however, does not occur evidently when the ageing proceeds further from the moment of 4 hours, which is quite different from that observed in the YS. The comparative observing suggests that the occurrence of recovery does not influence the TS. It is believed that the contribution of existing dislocations to the TS is much less to compare with that to the YS. The annihilation of dislocations, that developed in the later stage of the aging, then contributes much more to the decrease of YS that to that of TS.

Comment 4-4 How many samples have been tested? What is the confidence interval for the values obtained when testing the specimens in Figure 5?

Comment 4-5 What is the reproducibility of the results? Or were the tests carried out on one sample? Confidence intervals should be plotted in Figures 6 a-c, as well as information on Fig. 6d should be entered in the figure caption.

Reply to comment 4-4 and comment 4-5

  The number of sample used in the tensile testing have been added into the revised version. Also, the confidence interval that obtained in the testing has also been added into the revised manuscript.

The corresponding revisions are as follows—

All these specimens were then put into the tensile testing on an Instron 5569 tensile testing system at a strain rate of 2mm/min, referring the standard of GB228-87. Three specimens were used for the tensile testing for each condition and the averaged data were used for evaluating the mechanical properties of specimens.

Round 2

Reviewer 1 Report

Please recheck english language in some of the revised sections. There are minor grammar related errors. 

I would in the conclusions also point out the difference in character of the aging precipitation between stretched (mainly theta') and non stretched (mix of mainly theta" and minor theta' (at best)

Author Response

Thanks a lot for the reviewers’ professional comments on the manuscript that we submitted to the journal of Materials (No. Materials_1136699). The authors have revised the manuscript carefully regarding on the comments.

Comments 1-1

Please recheck english language in some of the revised sections. There are minor grammar related errors. 

Reply to comment 1-1

The english language has been checked carefully all through the manuscript to exclude the grammar related errors. 

Comments 1-2

I would in the conclusions also point out the difference in character of the aging precipitation between stretched (mainly theta') and non stretched (mix of mainly theta" and minor theta' (at best)

Reply to comment 1-2

 Thanks for the suggestion. The description of the precipitates in stretched/non-stretched specimens was added into the conclusions.

The corresponding revisions are as follows—

The precipitation of G.P. zones and the transition to θ″ shifts to lower temperature when as-quenched Al-Cu alloy sheet is stretched for more than 5%. The high density of dislocations developed during the stretching contributed to the acceleration in precipitation. Precipitates in stretched specimens are mainly θ′-phase while mixed precipitates (mainly θ″ and minor θ′) developed in non-stretched specimens.

Reviewer 3 Report

The article focuses on behavior of 2219 Al alloy ageing at two different level of temperature and 3 level of pre-stretch up to 15%. The paper contains major improvements and can be published in the present form. 

Author Response

Thanks a lot for the reviewers’ professional comments on the manuscript that we submitted to the journal of Materials (No. Materials_1136699). The authors have revised the manuscript carefully regarding on the comments.

Comments 3-1

The article focuses on behavior of 2219 Al alloy ageing at two different level of temperature and 3 level of pre-stretch up to 15%. The paper contains major improvements and can be published in the present form. 

Reply to comment 3-1

Thanks for the approval from the reviewer.

The manuscript has also been checked thoroughly to improve the presentation.
